# Structure and Dynamics of the EGF Receptor as Revealed by Experiments and Simulations and Its Relevance to Non-Small Cell Lung Cancer

**DOI:** 10.3390/cells8040316

**Published:** 2019-04-05

**Authors:** Marisa L. Martin-Fernandez, David T. Clarke, Selene K. Roberts, Laura C. Zanetti-Domingues, Francesco L. Gervasio

**Affiliations:** 1Central Laser Facility, Research Complex at Harwell, STFC Rutherford Appleton Laboratory, Harwell Oxford, Didcot, Oxford OX11 0QX, UK; dave.clarke@stfc.ac.uk (D.T.C.); selene.roberts@stfc.ac.uk (S.K.R.); laura.zanetti-domingues@stfc.ac.uk (L.C.Z.-D.); 2Department of Chemistry, University College London, London WC1H 0AJ, UK

**Keywords:** EGFR, lung cancer, receptor signaling, structure, MD simulations

## Abstract

The epidermal growth factor receptor (EGFR) is historically the prototypical receptor tyrosine kinase, being the first cloned and the first where the importance of ligand-induced dimer activation was ascertained. However, many years of structure determination has shown that EGFR is not completely understood. One challenge is that the many structure fragments stored at the PDB only provide a partial view because full-length proteins are flexible entities and dynamics play a key role in their functionality. Another challenge is the shortage of high-resolution data on functionally important higher-order complexes. Still, the interest in the structure/function relationships of EGFR remains unabated because of the crucial role played by oncogenic EGFR mutants in driving non-small cell lung cancer (NSCLC). Despite targeted therapies against EGFR setting a milestone in the treatment of this disease, ubiquitous drug resistance inevitably emerges after one year or so of treatment. The magnitude of the challenge has inspired novel strategies. Among these, the combination of multi-disciplinary experiments and molecular dynamic (MD) simulations have been pivotal in revealing the basic nature of EGFR monomers, dimers and multimers, and the structure-function relationships that underpin the mechanisms by which EGFR dysregulation contributes to the onset of NSCLC and resistance to treatment.

## 1. Introduction

This review aims to discuss EGFR structure/function mechanisms drawn from high-resolution experimental and theoretical results. The sources are mostly X-ray crystallography combined with dynamical insights from atomistic MD simulations, but also include results from single molecule experiments predicting structural-function relationships testable by MD simulations. Placed in the context of structural results available for wild type and mutant EGFR, we also discuss mechanisms leading to the onset of dysregulated cell growth in NSCLC and resistance to targeted therapies. 

### 1.1. EGFR and Its Connection with Oncogenic Cell Growth

The discovery of EGFR lagged ~25 years behind that of its first cognate ligand, the epidermal growth factor (EGF). EGF is a polypeptide isolated in the late 1950s from the mouse submaxillary salivary gland that enhances epidermal growth and keratinization (reviewed in [1]). A single pass 170 KDa transmembrane glycoprotein, the EGFR was the first cell surface receptor to be found to exhibit tyrosine kinase activity [2]. EGFR (aka HER1/ErbB1) is also the founding member of the human EGFR (HER) tyrosine kinase family (HER2/ErbB2/Neu, HER3/ErbB3, and HER4/ErbB4), four receptors that are among the ~60 receptor tyrosine kinases in the human genome [3]. Evolved from one receptor/ligand pair in nematode *Caenorhabditis elegans* through one receptor with multiple ligands in *Drosophila melanogaster* [4,5], the human (and vertebrate) EGFR family detects signals from 13 cognate polypeptide growth factor ligands. Of these, 7 EGFR-binding agonists—namely EGF, transforming growth factor α (TGFα), amphiregulin (AREG), betacellulin (BTC), epigen (EPN), epiregulin (EPR) and heparin-binding EGF-like growth factor (HB-EGF)—contain a characteristic EGF-like consensus domain responsible for EGFR binding, which involves six spatially conserved cysteine residues that form three intermolecular disulphide bonds [6,7]. All EGFR ligands are transcribed as transmembrane precursor proteins and are subsequently cleaved at the plasma membrane by cell surface proteases in order to release mature, active growth factors competent to bind EGFR (reviewed in [8]). Different EGFR binders display high affinity (EGF, TGFα, BTC, and HB-EGF) or 10- to 100-fold lower affinity (EREG, EPGN, and AREG) [6], and are capable of eliciting distinct extracellular conformations in EGFR [9], and thereby different cellular responses (for a review, see e.g., [10]).

The discovery that the dysregulated activation of EGFR is fundamentally important in cell transformation was made when the complete amino acid sequence encoding the human EGFR, which was derived in 1984 from cDNA clones extracted from placental and A431 carcinoma cells [11], revealed a high degree of homology with the v-erbB mRNA retroviral oncogene of the avian erythroblastosis virus, which encodes a truncated homologue of the human EGFR [12]. These results highlighted the importance of tyrosine phosphorylation and EGFR signalling in cancer biology. A substantial body of subsequent work has since shown that EGFR is frequently hyperactivated in human cancers via mutation and/or overexpression [13]. Of specific relevance in this review are the somatic mutations of EGFR that are associated with the development of lung cancer. This cancer type is not only the most common cause of cancer deaths worldwide in the past few decades (reviewed in [14]) (e.g., 1.6 million related deaths worldwide were reported in 2012 [15]) but also displays a five-year survival rate (~17.8%) much lower than that of other leading cancers [16]. Treatment has evolved from surgery combined with cisplatin-based chemotherapy to the use of personalised targeted therapies. This approach is exemplified in NSCLC, a group of histological diverse subtypes found in more than 80% of all lung cancer patients [14], in which tumour-driving *EGFR* mutations occur in approximately 10–20% of cases [17,18]. Mutant EGFR-driven NSCLC tumours are nowadays the best-studied example of oncogenic addiction in lung cancer [19]. The era of targeted therapies begun in NSCLC with the deployment of the first compounds designed to switch off specific signalling molecules, with a strong focus on aberrant, constitutively active EGFR mutant moieties to arrest oncogenic EGFR-dependent uncontrolled cell proliferation [17,18,20,21,22]. Subsequently, other gene alterations, including anaplastic lymphoma kinase gene (*ALK*) rearrangements, c-ros oncogene 1 (*ROS1*) fusions and B-Raf proto-oncogene (*BRAF*) mutations have similarly led to the development of additional targeted therapeutics [23]. However, ubiquitous drug resistance inevitably emerges after one year or so of treatment with targeted drugs. 

Another avenue of intervention has been to target molecular chaperones, like the heat shock protein HSP90 [24,25,26]. This chaperone has a fundamental role in regulating kinase activity, stabilizing and promoting the multimerization of numerous kinases, including members of the EGFR family, some of which (such as ErbB2) are obligatory clients of Hsp90 and are rapidly degraded upon its inhibition [27]. What is more, the increased expression of Hsp90 allows cancer cells to tolerate adverse environments and internal alterations deriving from accumulated mutations that would otherwise be lethal [28,29]. 

### 1.2. A Brief Description of Methods to Derive EGFR Structural Properties

A number of methods have been employed to determine or infer the structure of EGFR, either purified or within the cellular environment, to improve our understanding of the mechanisms driving drug resistance and to suggest the means to bypass it. The most commonly applied method for rational drug design is x-ray crystallography. This method almost requires no introduction, given its familiarity in structural biology, but in short, the protein of interest is first purified and then crystallized. The crystal is then placed in an x-ray beam, produced either by an x-ray generator or a synchrotron, and an electron density map determined from the measured intensities of the crystal’s diffraction pattern. This brief description brushes over decades of development of this technique, across the areas of protein crystallization, x-ray source and detector development, and data analysis methods. Those wishing to read more of this story are referred to, for example, a recent methods compendium [30]. X-ray crystallography now regularly delivers protein structures with sub-Angstrom resolution and has been hugely influential in biological research, including in studies of EGFR [31,32,33,34,35,36,37]. The reach of this technique is demonstrated by the fact that, at the time of writing, the RCSB Protein Data Bank [38] contains more than 133,000 protein structures determined using x-ray crystallography.

Despite the immense power of x-ray crystallography, it still by itself has not provided a universal solution to the impinging questions of how changes in protein structure underpin drug efficacy and resistance. One obvious consideration is that it requires proteins to be crystallised, and many multi-domain or flexible proteins, like EGFR, have only proved to be amenable to crystallisation fragment by fragment, in complex with exogenous binders, to stabilise a conformation, and devoid of post-translational modifications. What is more, crystal contacts, expression tags and crystallisation conditions might over-stabilize specific conformations or destabilise functionally-important states [39]. Often flexible regulatory regions are not visible. Methods are available that can be used to determine protein structure and dynamics in solution. One of these is nuclear magnetic resonance spectroscopy (NMR), in which the protein (either in solution or in the solid state) is placed in a strong magnetic field and exposed to a radio frequency (RF) field. The absorption of RF by the nuclei of atoms in the protein is sensitive to the environment of the atom, and reports on the presence of adjacent nuclei. It is, therefore, possible to build up a map of distances between the nuclei, which can then be used to determine the protein’s structure. Advanced NMR techniques are also able to reconstruct protein dynamics across different time-scales and can be applied to molecules in solution or the solid state. In solution NMR, spectral transitions are sharp, due to averaging of anisotropic NMR interactions. Solid-state NMR (ssNMR) spectra are broader, as the full effects of anisotropic or orientation-dependent interactions are observed in the spectrum. ssNMR methods are particularly applicable to membrane proteins, because these proteins are immobilized in the lipid bilayer on the timescales of the chemical shift and dipolar coupling spin interactions [40]. However, NMR also has a blind spot that makes interesting regulatory regions invisible due to signal broadening. Again, NMR is a complex and highly developed technique, and for more information, the reader is referred to the literature, e.g., [41]. 

Another method that can provide useful insights in the structure of uncrystallised proteins and protein complexes is mass spectroscopy (MS), which measures the mass-to-charge ratio of ions to identify and quantify molecules in simple and complex mixtures. In recent years it has developed into an important tool for the study of protein structure, particularly with the introduction of methods such as matrix-assisted laser desorption ionization (MALDI) and electrospray ionization (ESI) [42]. In the proteomics field, MS can, for example, provide information on protein complex topology, protein folding, folding interactions and detect specific post-translational modifications in complex biological mixtures. 

Small angle X-ray solution scattering (SAXS) is another method that can provide protein structural information albeit at low resolution. It works by measuring the intensity of elastically scattered x-rays from protein molecules in solution. The scattering pattern provides information on the shape and size of the protein being investigated. SAXS is often combined with atomic resolution structures to enable modelling of protein complexes [43].

Lastly, transmission electron microscopy (TEM), and specifically single particle TEM under cryogenic conditions is nowadays becoming the method of choice to investigate atomic resolution protein structure in non-crystallised purified protein samples. In this technique, a solution of the protein is flash-frozen and imaged in the TEM in a thin layer of amorphous ice. Many (up to tens of thousands) of images are recorded of individual protein molecules, across a wide range of orientations. These are used to computationally reconstruct a model of the protein, at high resolution. Advances in detector technology and analysis software mean that this technique can now deliver protein structures at resolutions approaching that of x-ray crystallography [44].

The above techniques are currently used to resolve atomistic structures of isolated, purified proteins either in crystalline form or in solution. However, in order to fully understand the protein’s function in biological systems, it would be desirable to measure protein structure in the cell. Currently no technique exists that can provide atomic resolution structure in cellulo, but a number of methods are used to obtain lower-resolution structural information, which can be correlated with high-resolution structure of purified proteins, and models from MD simulations, to provide a detailed structural picture inside the cell (see for example [45]). One such method is Förster resonance energy transfer (FRET), in which two protein types of interest are labelled with a fluorescent marker, one with the so-called “donor”, and the other with an “acceptor”. The markers are chosen so that the fluorescence emission spectrum of the donor overlaps the absorption spectrum of the acceptor. If the molecules are close enough (< ~8 nm), on absorption of a photon the donor can transfer its energy non-radiatively to a nearby acceptor. The efficiency of this FRET process varies with the 6th power of the separation of the molecules, so provides a very precise (sub-nm) measurement of inter-molecular separation in the sub-8 nm range [46]. The efficiency of FRET can be determined by measuring the fluorescence lifetime of the donor molecule; this can be done in a microscope using a technique known as fluorescence lifetime imaging microscopy (FLIM), allowing particular areas of the cell to be investigated [47].

FRET is a powerful technique for measuring relatively short intermolecular distances, but larger complexes of molecules require a different approach. A method has been developed that makes use of the principle of single-molecule localization, by which the location of a single fluorescence emitter can be determined with nanometre precision [48]. The technique of fluorescence localization imaging with photobleaching (FLImP) measures the shift in position of fluorescence from a pair of emitters when one of the molecules photobleaches. This enables the distance between the molecules to be determined in the range from ~5 nm to 60 nm. By accumulating many of these distance measurements it is possible to determine the distribution of distances within protein complexes, providing, for example, the proportion of dimers, trimers, and higher-order oligomers and their architecture on cells. FLImP has been used to investigate EGFR oligomers in cells under a range of conditions [49,50].

No single structural technique described above is sufficient to fully characterize the structure and organization of molecules in the cell. However, the combination of atomic resolution structures of purified proteins and lower-resolution measurements of protein complexes in the cellular environment is very powerful, particularly when the information is combined with modelling and MD simulations. 

### 1.3. MD Simulation Methods Applied to EGFR Research

The MD simulation methodology, developed in the 60s, was first used to simulate the dynamics of a small protein in 1977 [51] and has since become a suite of established techniques to investigate structure-function relationships. As the available computer power increased exponentially, MD simulations have grown in size (from a few hundred to millions of atoms) and length (from ps to μs), thereby becoming a viable approach to study the conformational dynamics of many complex biomolecular systems and their functional implication. MD simulations nowadays play a fundamental role in understanding allosteric mechanisms like those underpinning EGFR signalling (reviewed in [52]). Atomistic MD simulations are based on the integration of Newton’s equation of motion and a simplified representation of the potential energy function, the force field. The most commonly used biomolecular force fields, such as Amber, CHARMM and OPLS force-field families, build on Lifson’s seminal work in the 60s, approximate the potential energy function by a combination of harmonic springs for bond lengths and angles, a cosine expansion for dihedrals, Lennard-Jones potentials for non-bonded interactions, and point charges and Coulomb’s law for electrostatic interactions (Lifson type force fields). Despite their simplified form, biomolecular force fields for water, proteins, glycans, lipids, DNA and RNA, have been systematically refined and their most recent iterations (such as Amber14, Amber-disp and CHARMM36m) are able to reproduce the structural, dynamical and thermodynamical properties of very complex systems with surprising accuracy [53,54,55,56,57,58,59,60]. However, the time-scales accessible to MD simulations are still limited by the small integration step used to evolve Newton’s equations, which must be comparable to the characteristic timescales of the fastest molecular motion (10^−15^ s for bonds vibration). Significant structural rearrangements, as the transitions between different kinase conformations, take place in time scales of tens to hundreds of microseconds and are difficult to observe in a typical MD simulation that, even using the fastest MD code on a modern supercomputer might take a month to sample a few μs. Only recently, thanks to algorithmic and hardware advancements, including supercomputers such as ‘Anton’ [61], a special-purpose system for MD simulations using a large number of application-specific integrated circuits (ASICs), MD codes running on GPUs and enhanced-sampling algorithms [62,63,64,65,66], the time-scales needed to sample these phenomena have become accessible. Using Anton, it was possible to observe large scale conformational changes in EGFR (e.g., [49,67]). But even on Anton, only one or a few reactive events can be observed. To quantify the free energy and population differences between conformers, allowing a direct comparison with experiments, a coarse-grained model or a more efficient sampling approach is needed [68]. To this end, algorithms that combine a large number of trajectories, *accelerate rare events* and compute the free energies can be used [62,64,65,66,68,69,70,71]. In the case of kinases, free energy methods have been used to reconstruct the conformational free energy profile along a relevant coordinate (collective variable or CV) that approximates the reaction coordinates [72,73]. However, both the choice of a CV describing all relevant conformational changes and the convergence of the reconstructed free energy profile are significant concerns [74] and may lead to a simplistic description that is difficult to reconcile with the most up-to-date NMR evidence. A possible solution is to use “CV-free” algorithms that enhance the sampling by using temperature, such as parallel tempering, a modified Hamiltonian (as in Hamiltonian replica exchange) or combine many trajectories with Markov state models [75]. CV algorithms can also be combined with multiple replica algorithms to improve convergence, as in the case of parallel tempering metadynamics (PT-metaD) [76]. PT-metaD was extensively and successfully used to calculate the conformational free energy landscape of the catalytic domains of Abl (Abelson murine leukemia viral oncogene), Src, originally identified by homology to the Rous sarcoma virus oncogene protein pp60 (v-src), fibroblast growth factor receptor (FGFR), and P38 protein kinases, and validated by NMR, Hydrogen deuterium exchange (HDX) mass spectrometry, and FRET experiments [39,77,78,79]. PT-metaD was also used to clarify the effect of common oncogenic mutations on the conformational free energy landscape of the catalytic domains of Abl, EGFR and BRAF [80,81,82]. 

## 2. The Glycosylated EGFR Monomer and Interactions with the Supporting Bilayer 

The human EGFR gene encodes a 1210 amino acid sequence, in which the first 24 correspond to a putative signal peptide. (The amino acid numbering used in this review does not count these). EGFR has a single-pass transmembrane (TM) α-helical region (residues 618–44) embedded in the plasma membrane that connects the N-terminal growth factor ligand-binding extracellular module (ECM) (residues 1–617) to its associated intracellular module (ICM) (Figure 1a). The ICM is made of a juxtamembrane (JM) domain (residues 645–677), composed of a N-terminal portion (JMA) (residues 645-663) and a C-terminal portion (JMB) (residues 664–677), a kinase domain (residues 678–954), and a long and disordered carboxy-terminal tail region (residues 955–1186). This overall topology has been conserved through evolution reflecting EGFR’s critical signalling role in cells and tissues

### 2.1. The Ligand-Binding ECM Monomer

The ECM of EGFR is made of four subdomains (DI-DIV), of which DI displays a similar topology to DIII, and DII and DIV are cysteine-rich domains [31,83]. Its main functional role is to bind growth factor molecules in the extracellular milieu. The main growth factor binding site is DIII. This was deciphered 30 years ago in an elegant study in which various domains of the chicken EGFR were substituted by domains of the human EGFR to generate chimeric chicken/human receptors [84]. Because the human EGFR shows a ~100-fold higher binding affinity towards EGF, variations in the affinity of EGF for different chimeric receptors identified DIII as a major EGF binding domain. 

In the first published X-ray structure of the nearly complete solubilised monomeric ECM of EGFR, often referred to as sEGFR, the ECM was weakly bound to EGF at DI with very low affinity (at low pH) [85]. The data revealed a folded-over structure, so-called the tethered conformation, in which a β-hairpin from DII forms an interaction with DIV (Figure 1b). This tethered conformation was later found for the solubilised ECMs of two other members of the EGFR family (HER3 through SAXS and HER4 in a crystal [86,87]), for EGFR bound at DIII with the Fab fragment of cetuximab/Erbitux/IMC-C225 [88] or a Nanobody/VHH domain (Figure 1c) [89]. Because the β-hairpin of DII is the major dimerisation site previously revealed by ligand-bound dimer structures (discussed later in Section 4.1), the tethered conformation was proposed to be an autoinhibitory mechanism preventing the spontaneous formation of ligand-independent EGFR dimers (reviewed in [90]). However, questioning this hypothesis, quantitative ligand-binding experiments on cells only found a limited role for the DII-DIV tether in EGFR function (e.g., [87,90,91,92]).

It is conceivable that receptors on the cell surface can sample a wider conformational landscape than that reported by crystal structures because the latter originate from highly concentrated homogeneous preparations of purified fragments. Indeed, recent ssNMR-based experiments on EGFR-rich A431 cell membrane vesicles confirmed that the ligand-free state of the full-length EGFR monomer is highly dynamic and that the ability to explore different conformations is of critical importance for EGFR function [95] MD simulations can explore the wider conformational landscape at atomic resolution to report conformational changes and alternative ECM conformations. For example, using Anton-based technology Arkhipov et al. [67] carried out long-timescale atomistic simulations starting from the tethered ECM monomer structure before and after removal of the bound EGF molecule [85]. The results showed that, consistent with a limited role for the tether, DII and DIV disengaged in the scale of microseconds, breaking the tether interaction. To explore further the conformational landscape, Arkhipov et al. started from a subunit of the ligand-bound dimer structure in which monomer ECMs adopt an extended conformation [96] (Figure 1d). The results again revealed a significant conformational rearrangement after 1–5 μs, by which DIV bends around a ‘hinge’ (residues 502–514) displacing the C-terminus of DIV towards the dimerisation arm of DII, the endpoint resembling a compact hybrid between the starting extended conformation and the tethered conformation (Figure 1e). The results of these simulations are consistent with a high degree of ECM monomer flexibility [95].

### 2.2. Glycosylation Regulates the Structure and Function of the ECM 

Another difference between crystal structures and data from cells is that the ECM of cell surface EGFR is heavily glycosylated with sugar moieties making up nearly 25% of the 180 KDa net mass of the receptor. Glycosylation is fundamental to many receptor functions, such as ligand-independent activation, growth factor binding affinity, and receptor dimerisation [97,98,99]. Glycosylation also undergoes profound alterations in cancer, including in NSCLC [100,101,102,103].

A prevalent form is N-glycosylation, which is the covalent binding of sugar moieties to the amide nitrogen atom of an asparagine [104]. Mass spectrometry, biochemical and activity assays, and high-throughput methods are among the methods that have been used to investigate the glycosylation sites of the full-length membrane-bound EGFR (see for example [105]). As shown in Figure 1d, results revealed 10 Asparagines located within consensus *N*-glycosylation sequences (Asn-X-Ser/Thr, where X can be any amino acid except proline [11]). These are Asn104, Asn151 in DI, Asn172 in DII, Asn328, Asn337, Asn389, and Asn420 in DIII, and Asn504, Asn544, Asn579 and Asn599 in DIV. Of the 10 Asparagines, nine are glycosylated (i.e. all except Asn172) [106,107,108,109]. However, suggesting that the consensus sequence is not necessary for N-glycosylation, a non-consensus asparagine, Asn32 in DI, which forms part of an atypical glycosylation sequence (Asn-Asn-Cys), was also reported to be both N-glycosylated and fucosylated (addition of fucose to the oligosaccharide glycan) [110]. 

The amino acid position of the sugar moieties reflects the breadth of different EGFR functions regulated by glycosylation. For example, using a cell line expressing a point mutant of EGFR (N579Q-EGFR), the glycosylation of Asn579, located in DIV at the tip of the tether loop with DII, was shown to reinforce the tether interaction between these domains [111]. Crystal structures do not typically include large chemical modifications, like glycosylation, because glycans are highly flexible [104], and introduce microheterogeneity into the protein, both detrimental to crystal growth. Because of these challenges, MD simulations have played a key role in revealing the mechanisms by which *N*-glycan moieties regulate EGFR structure. For example, Taylor et al. [105] deployed all-atom MD simulations starting, like Arkhipov et al. [67], from a monomer subunit of the ligand-bound dimer structure [31] to characterize the conformational preferences of the ECM in the presence and absence of growth factor and *N*-glycosylation. The results show that the presence of ligand confers the major conformational stability to EGFR and that this followed by *N*-glycosylation and lastly dimerisation, which reveals the hierarchy of the structural variables determining the extracellular conformation of EGFR.

Atomistic MD simulations were also performed by Irani et al. [112] to investigate how *N*-glycosylation regulates ligand-binding affinity. The simulations showed that *N*-glycosylation results in the formation of noncovalent interactions between glycans and amino acids nearby the EGF binding site. Such interactions result in stronger electrostatic interactions between the growth factor ligand and EGFR that stabilize the ligand-binding site, explaining the molecular mechanism by which *N*-glycosylation regulates growth factor affinity. 

### 2.3. Linking Across the Lipid Bilayer to the ICM

The flexibility of EGFR compounded with methodological limitations related to the crystallisation of membrane proteins have so far prevented the crystallisation of the full-length receptor. Using Anton, Arkhipov et al. [67] carried out a 5 μs-long simulation of the near-complete EGFR molecule embedded in a lipid bilayer made of 1-Palmitoyl-2-oleoyl-*SN*-glycero-3-phosphocholine/phosphatidylserine (POPC/POPS) (Figure 1f). The simulated unit included the ECM, the TM helix, the JMA segment, which becomes embedded in the bilayer, the JMB portion, and the kinase domain in its inactive conformation (discussed later in Section 3.2). The kinase domain was oriented relative to the bilayer to enable contacts between the negatively charged membrane and two positively charged surface regions of the kinase domain (Lys698, Lys690, Lys692 and Lys715, and Arg779, Arg817, Lys851 and Lys889). Because the kinase domain was sequestered by the membrane, it was proposed that these ionic interactions might contribute to the autoinhibition of the kinase active state because the membrane would occlude the substrate-binding site of the kinase [67]. 

The relaxed ECM monomer established a large interfacial contact with the supporting lipid bilayer, which predicts a very short separation between its N-terminus and the outer leaflet of the membrane. This seemed inconsistent with the larger distances reported from FRET measurements of glycosylated intact receptors on cells [113,114,115]. To address this, Kaszuba et al. [94] considered whether the differences could be attributed to N-glycosylation by simulating a near-full-length glycosylated receptor (residues 2–994). Atomistic MD simulations were carried out on both the nonglycosylated receptor and after attaching to the ECM in silico universal *N*-glycan core Man_3_GlcNAc_2_ residues at positions Asn151 (DI), Asn172 (DII) and Asn389 together with Asn420 (DIII) (Figure 1g,h). Instead of the binary lipid chain used by Arkhipov et al. [67], the reconstructed chain was embedded in a ternary lipid bilayer (1,2-dioleoylsn-glycero-3-phosphocholine (DOPC)/ sphingomyelin (SM)/Cholesterol) designed to mimic a real mammalian plasma membrane. Earlier experimental work in EGFR proteoliposomes show this lipid composition to be critical to prevent ligand-independent kinase activation [116]. The results from these simulations, which covered ~1 μs, revealed that the presence of the Man_3_GlcNAc_2_ glycan residues significantly altered the relative arrangement of individual domains of the ECM and their alignment on the membrane. The *N*-glycosylated sites were found to act as ‘molecular cushions’ lifting DI and DIII from the membrane so that DIII no longer made contact with the membrane [94].

## 3. Kinase Domain Conformations and Their Coupling to Tyrosine Kinase Inhibitors (TKIs) 

### 3.1. Topology of the Catalytic Kinase Domain 

Protein kinase domains are found in 2% of eukaryotic genes. Their function is to catalyse the transfer of a γ-phosphate (PO_4_^3−^) of adenosine triphosphate (ATP) to the hydroxyl-group of a tyrosine in the target substrate, a posttranslational modification that emerged more than a billion years ago in single-cell organisms [117]. Because of the critical importance of tyrosine phosphorylation to intracellular communication [3,118], tyrosine kinase structure is extremely well conserved [119]. Kinase domains consist of a smaller N-lobe and a larger C-lobe [120] (Figure 2a). The N-lobe mostly comprises five beta sheet strands (β1−β5) and a conserved alpha helix (αC) (residues 729–744). The *C*-lobe is mostly composed of five alpha helices (αE, αF, αG, αH, αI). The ATP-binding site is within a deep cleft sandwiched between the N-lobe and C-lobe and underneath a highly conserved glycine-rich phosphate-binding loop, which connects β1 and β2 in the N-lobe. The glycine-rich loop coordinates closely with the phosphates of ATP via backbone interactions [121]. The αC helix contains an absolutely conserved glutamate (Glu738 in EGFR), which in the active state establishes an ion pair with a conserved lysine residue in β3 (Lys721 in EGFR) coordinating the α and β phosphates of ATP (Figure 2b). The C-lobe borders the ATP-binding cleft from below, contributing a highly conserved catalytic loop (Asp812-Asn818 in EGFR), within which Asp812 interacts with the attacking hydroxyl site chain of the tyrosine substrate, while Asn818 forms hydrogen bond interactions that orient Asp812. The C-lobe also contributes the key regulatory activation loop (A-loop) (Asp831-Val852 in EGFR), which contains at its base a conserved Asp-Phe-Gly (DFG) motif (Asp831-Gly833 in EGFR). 

The C-terminal tail is EGFR’s autocatalytic substrate (Figure 1a). It binds the kinase in an extended conformation across the front end of the ATP binding pocket close to the γ-phosphate of the nucleotide [122]. Interactions between the kinase and its substrate are supported by the overall conformation of the A-loop, including a β9 strand platform. The auto-catalytic activity of EGFR results in the phosphorylation of main tyrosine residues along its C-terminal domain (Tyr974, Tyr992, Tyr1048, Tyr1068, Tyr1086, Tyr1101 and Tyr1173) [123]). C-terminal tail phosphotyrosines serve as docking sites for the recruitment of signalling proteins and adaptors containing Src homology 2 (SH2) and phosphotyrosine-binding (PTB) domains [4]. Tyrosine phosphorylation is immediately responsible for the downstream signalling pathways engaged by EGFR via this fashion, namely the RAS-RAF-MEK-ERK pathway and the PI3K-AKT-mTOR pathway, promoting cell proliferation and cell survival, respectively [124]. 

### 3.2. TKIs Bind the ATP-Binding Kinase Pocket and Stabilise Active and Inactive Conformations

TKIs, as those shown bound to EGFR and ABL tyrosine kinases in Figure 2a–e, are small ATP-mimetic molecules that block phosphorylation by competing with ATP [132]. TKI binding is facilitated by the structural flexibility displayed by the glycine-rich loop in the absence of ATP [133] (Figure 2a). To date, seven EGFR family-targeted TKIs have been FDA-approved [134], including gefitinib (IressaTM) and erlotinib (TarcevaTM), neratinib (NerlynxTM), afatinib (GilotrifTM), lapatinib (TykerbTM) and osimertinib (TagrissoTM) and two multi-kinase (including EGFR) targeting drugs brigatinib (AlunbrigTM) and vandetanib (CaprelsaTM). TKIs have played a critical role in the fight against NSCLC by harbouring activating mutations in the EGFR kinase. Recent developments in multi-targeted drug discovery have also produced multi-targeted TKIs such as foretinib, which can target both c-Met and AXL [135], two of the main bypass pathways of EGFR. Combination therapy with trastuzumab [136], with the insulin-like growth factor receptor 1 (IGF1R) TKI linsitinib [137] or with the anti-HER3 antibodies currently in development [138] can also prove beneficial to shut down compensatory signalling. Other new therapies instead reverse the immune tolerance towards cancer through the blockade of negative immune regulators such as PD-1/PD-1L and CTLA4 have enjoyed vast success in clinical trials of NSCLC and bypass the need of targeting EGFR specifically, providing an attractive therapeutic [139,140].

TKIs are classified according to the structure of their drug-enzyme complexes (reviewed in [141]). The observed differences between type I and type II binding modes is often due to the para-substitution on the aniline ring (Figure 2c). Type I usually carry a leading halogen whereas type II bear a bulky group (a benzyl ring) [125]. Consequently, type I inhibitors bind the ATP pocket but do not penetrate the allosteric pocket. Type II create an additional cavity deep into the ATP-binding cleft to accommodate their leading benzyl ring. This cavity requires a large distortion in the glycine-rich loop achieved by interactions between the benzyl ring of the TKI and a conserved aromatic residue [142]. In many tyrosine kinases, contacts between type II inhibitors with the adjacent allosteric site are facilitated by a kinase conformation in which the DFG motif at the N-terminus of the A-loop flips away from the catalytic centre (DFG-out conformation) (Figure 2d). 

TKIs are also classified according to their mode of binding [143,144,145]. The first generation binds reversibly to the target enzyme and interrupt signalling by outcompeting ATP [146]. The second generation integrates as a reactive warhead moiety the acrylamide fragment that alkylates a conserved cysteine residue in the kinase domain (Cys773) [147], thereby establishing an irreversible bond. The third and fourth generations are allosteric, EGFR mutant-specific and designed to overcome resistance to TKIs in NSCLC (discussed further below in Section 3.3). 

TKI binding stabilises different conformations of the kinase domain and this effect has been exploited to facilitate crystallisation. Indeed, the first structures of EGFR’s kinase domain revealed by X-ray crystallography were stabilised in an active conformation via the binding of a first generation type I TKI [120] (Figure 2a). The data showed structural features characteristic of the kinase active conformation [133]. These include: (i) the αC-helix is closely packed against the body of the N-lobe (αC-in), thereby orienting Glu738 with respect to Lys721 to form the catalytically important salt bridge that couples the conformation of the αC-helix to nucleotide binding (Figure 2b); (ii) the DFG motif is in contact with the catalytic site (DFG-in) (Figure 2d), thus underpinning ATP coordination; and (iii) the A-loop is opened and arranged to bind substrate (Figure 2a). 

In turn, the first structure revealing an inactive EGFR kinase conformation was bound to the first generation TKI lapatinib [125]. Typical of inactive kinases, this structure displayed an ‘αC-out’ conformation (Figure 2d), in which the αC-helix is displaced outwards swinging out the conserved Glu738, thus breaking the salt bridge with Lys721, and the A-loop is collapsed against the catalytic site, thus occluding both nucleotide and substrate binding. However, the DFG motif pointed towards the ATP-binding site, thus displaying a DFG-in conformation (Figure 2e). At the time, the finding of the DGF-in conformation in the presence of lapatinib was unexpected, given that the ability to adopt the DFG-out conformation contributes to the increased selectivity of type II inhibitors versus type I [148]. For this reason, lapatinib is sometimes classified as a type I½ inhibitor [141]. In the DFG-in conformation, residues L834 to D838 of the A-loop, which form the β9 strand in the active conformation, also form a two-turn helical segment that packs against the shifted αC helix. Since the inactive conformation of EGFR’s kinase resembles the inactive conformations of Src and CDK2 kinases [126], it is often referred to as the ‘‘Src/CDK2-like inactive”. The DGF-out conformation was later found for EGFR’s kinase domain, for example, in the crystal structure of the complex between the EGFR kinase domain displaying the inactivating V924R mutation and an inhibitory mitogen-inducible gene 6 (MIG6) peptide [37], a negative regulator of EGF receptor-mediated skin morphogenesis and tumour formation, also known as RALT or Gene 33 [149]. 

Almost 60 PDB entries of EGFR kinase structures are nowadays available (reviewed in [150]). The analysis of this ‘experimental ensemble’ reveals the clusters of active and inactive conformers that dominate the wild type kinase native state. An unusual feature of EGFR’s kinase is that the structural coupling between the αC helix and the A-loop [133], which is essential for the adoption of the active kinase conformation, does not depend on the phosphorylation of the A-loop. In EGFR kinase this position is occupied by the highly conserved, functionally important Tyr845. In many other tyrosine kinases, e.g., the insulin receptor [3], a tyrosine in an equivalent position would stabilise the A-loop in the inactive conformation and would need to be trans-phosphorylated by a partner kinase to release the cis-autoinhibitory interactions and relax into an active conformation. 

### 3.3. NSCLC Oncogenic Mutations and Their Impact on Structure and TKI Interactions

Common somatic mutations relevant to NSCLC are located within *EGFR* exons 18-21, which encode a critical portion of the kinase domain [151], the most frequent is located in the A-loop (L834R). L834R-EGFR displays sensitivity to first generation TKIs [134], but the development of a dominant secondary amino acid substitution at Thr766 confers resistance to TKIs and unavoidably limits the long-term efficiency of these drugs [152]. The T766M mutation is located in β5 of the N-lobe, in the so-called ‘gatekeeper’ position because it controls the access of TKIs to a deep hydrophobic pocket in the ATP binding site [153]. Given that the gatekeeper residue can regulate the ‘in’ versus ‘out’ conformations of the conserved DFG motif, the resistance to TKIs conferred by the T766M mutation was first predicted to be due to a steric clash between the larger substituting methionine amino acid on the gatekeeper side chain of T766M and the aniline moiety of first generation TKIs. This is the so-called ‘gatekeeper’ hypothesis [153]. However, Yun et al. [154] showed that the Met766 residue can shift to accommodate inhibitor binding, and proposed instead that resistance can be accounted for by the greatly increased ATP-binding affinity (~8-fold) for T766M-EGFR, results later validated by Yoshikawa et al. and Red-Brewer et al. [155,156]. 

The key role in resistance to TKIs of the secondary T766M mutation has focused investigations into the development of inhibitors that target this mutant kinase. Partial sensitivity is maintained with type II irreversible inhibitors like neratinib or afatinib [157]. However, the unrestrained potency of type II irreversible TKIs against wild type EGFR results in severe epithelium-based toxicity, reducing to impracticable levels the doses of drug that can be safely administered [158]. Secondary resistance to type II irreversible inhibitors acquired through T766M can be countered with mutant-selective third generation TKIs, such as osimertinib (AZD9291) [152]. Used in clinic, osimertinib is a monoanilinopyrimidine compound that incorporates a Michael acceptor group that forms a covalent bond with Cys773 located at the edge of the ATP-binding pocket. By binding irreversibly, this inhibitor overcomes the increased ATP affinity of the T766M-EGFR mutant, while largely sparing wild-type EGFR, thereby minimizing unwanted side effects at doses required for meaningful therapeutic intervention (20–240 mg) [159]. Unfortunately, the efficacy of osimertinib follows an identical pattern of activity followed by resistance as its predecessors, which occurs either through loss of primary and secondary EGFR mutations or through the acquisition of tertiary mutations (e.g., C773S) blocking the formation of the potency-conferring covalent bond [160]. 

Oncogenic and other mutations in the EGFR kinase region impact the conformational free-energy landscape [80] and dynamics of the enzyme [161], adding to the differential enrichment of each conformer population by allowing the kinase to sample additional conformations. From these structural fingerprints underpinning enzyme dysregulation and resistance to TKIs can be ascertained. All-atom MD simulations have provided a complementary view on how conformational intermediates underpin enzymatic function and inhibitor binding. Using Anton, it was possible to perform a multiple µs-long MD simulations to study the conformational “flip” of the conserved DFG motif in the ABL kinase between the ‘in’ and ‘out’ states [162]. This “flip” is relevant as it is common with type II inhibitors and is believed to play a role in the selectivity of imatinib, the first kinase inhibitor to be used in targeted anticancer therapy. In their study, the authors identified a new, potentially druggable state, and highlighted the role of DFG aspartate protonation in favouring different conformations. However, evidence from crystal structures of wild type EGFR [127,163], L834R and L834R/T766M mutant EGFR [161] have also suggested that sensitivity and resistance to TKIs is most likely due to a complex interplay of protein conformational dynamics, competition from ATP, and state of phosphorylation, rather than to the conformational specificity of the inhibitors per se.

Long MD simulations were also used to study the binding of the inhibitor lapatinib to EGFR [164], which led the authors to postulate the existence of a hitherto unknown conformation of the αC-helix from discrepancies in the calculated rate of association. The authors proposed that the slower binding rate of lapatinib was due to the existence of a previously unknown conformation assumed by the αC-helix, different from the “out” inactive conformation that binds lapatinib, and which the protein must assume in order for the drug to bind. What is more, they proposed a new explanation for the mode of action of the widespread mutation L834R, affecting the dimerization dynamics of EGFR despite being distant from the dimerization interface (discussed further in Section 4.2). Their simulations are consistent with a stabilization of the αC-helix “in” conformation by the mutation, which suppresses the disorder of the helix and stabilizes the dimer. 

Sutto et al. further investigated the effects of oncogenic mutations (L834R and T766M) on the conformational free-energy landscape of the EGFR kinase domain through extensive MD simulations, parallel tempering and metadynamics [80]. The conformational free-energy landscapes reconstructed by the simulations show that, while the wild type EGFR mostly exists in the inactive conformation in which the A-loop partially blocks the access to the catalytic cleft, the oncogenic mutants destabilize the inactive conformation in favour of the active conformation. What is more, the mode of action of the two mutations differ in interesting ways. The L834R mutation not only stabilizes the active conformation of the A-loop at the expense of the inactive conformation but also rigidifies the αC-helix that is involved in the dimerization interface, reconciling previously contrasting views [126,164]. The T766M “gatekeeper” mutant favors activation of the A-loop by stabilizing a hydrophobic cluster, explaining the increased affinity for ATP. However, it also leads to a less ordered dimerization interface, possibly destabilizing the dimer. What is more, the combination of T766M with L834R shows a positive epistatic interaction. 

Using Anton, Shan et al. [164] reported unbiased atomistic simulations of the EGFR kinase monomer in the time scale of 25 μs showing transitions between the active state and both the Src-like inactive and the DFG-out inactive states of EGFR kinase. These simulations identified a locally disordered intermediate state with characteristics of both the active and inactive states, in which the αC-helix is partially disordered and adopts the αC-out inactive conformation, while the A-loop remains in an active conformation and displaying a β9 strand (Figure 2f). The spontaneous transition from the active to the so-called “Src-like inactive” conformation of the wild type kinase funnelled across two different pathways, as also observed by Li et al. [165], who combined targeted MD, unbiased MD and Bayesian clustering. In 60 μs long all-atom MD simulations, Shan et al. [131] revealed that the same local disordered state appeared as an intermediate in transitions between the active state, the DFG-in, Src-like inactive, and DFG-out inactive states of the EGFR kinase. Intermediate conformations were accompanied by unfolding (or ‘cracking’) at the hinge region between the N-lobe and C-lobe, together with the extraordinary conformational flexibility of the A-loop (Figure 2g), which highlights the overall key role played in by the latter in the conformational dynamics of tyrosine kinases. 

Shukla et al. [166] investigated key structural intermediates along the transition path between the active and inactive states. Using Markov state models to explore the conformational landscape of the c-src tyrosine kinase, potentially druggable allosteric sites in the intermediates were identified. Allosteric inhibitors (so-called 4th generation inhibitors) are a new type of mutant-selective drug and were identified in a screen of ~2.5 million compounds against purified L834R/T766M kinase [143]. Crystal structures revealed that the compounds bind an allosteric site created by the displacement of the regulatory αC-helix in an inactive conformation of the kinase [143]. The affinity for the double mutant kinase was in the low nM range and for the wild type kinase >50 μM. The findings suggested that, in combination with anti-EGFR antibodies such as cetuximab, 4th generation inhibitors could overcome resistance to the triple L834R/T766M/C773 mutation, which is resistant to all current EGFR-targeted therapies. Recent MD simulations have investigated the structural basis underlying the binding of 4th generation inhibitors [167]. The results revealed that the conformational destabilization of the short helix that carries Leu834 in the wild type exposes the allosteric pocket, which is otherwise occluded by a set of sidechains including L834. 

The effect of mutations in the N-lobe of the kinase were investigated by Paladino et al. [168] who used a novel computational scheme based on the analysis of internal protein energetics and flexibility/rigidity to describe protein dynamics in order to analyse the active and inactive forms of the kinase domain of EGFR in the presence and absence of the G695S mutation. Suggesting a mechanism by which N-lobe mutations favours the active state, energy decomposition analysis of the EGFR kinase incorporating the G695S mutation and in the active conformation showed a remarkable increase in the contribution of the N-lobe residues to the stability of the kinase domain. This was still present but was somewhat less prominent in the inactive state [168,169].

## 4. Regulation of EGFR Enzymatic Activation by Ligand-Induced Dimerisation 

### 4.1. The Stable Heart-Shape of the Ligand-Bound Extracellular Dimer Module

Ligand binding to the ECM of EGFR promotes the formation of a dimeric receptor (Figure 3a). Our understanding of the ligand-bound structure of the extracellular portion of dimeric human EGFR is based on three X-ray crystal structures. The two structures first published in 2002 show a ligand-bound dimer lacking almost the entire DIV and bound to two ligands [31,83]. This structure denominated the back-to-back dimer because dimerisation was almost exclusively mediated by a β-hairpin of DII of the ECM, or the dimerisation arm. A more recent structure of the back-to-back dimer [96] includes a clear density in the parts of DIV missing in the previous two (Figure 3a), identifying a small C-terminal DIV dimerisation interface which gives the ligand-bound dimer structure the heart-shape appearance recapitulated by electron microscopy studies [171]. Comparison between the tethered monomer and the extended back-to-back dimer reveals the ∼130° conformational rotation of DI and DII with respect to DIII and DIV that would be induced by ligand-binding to transition between the two structures. Notably, the rotating domains (DI and DII), which are rigid β-helix/solenoid structures, would not change conformation during the rotation. The putative orientation of DIV with respect to the membrane assumed in the cartoon of Figure 3a would also be largely maintained [85,169]. Ligand-binding would induce a different conformational transition upon binding the ECM suggested by MD simulations (Figure 1f). Reminiscent of the extension of a swinging arm, DI, DII and DIII would rotate clockwise around its hinge with DIV while DIV rotates anticlockwise, thus arriving at the extended conformation observed in the back-to-back dimer structure. 

Unlike the results from the MD simulations of the monomer ECM, for which the tethered structure conformation was unstable during MD simulations, Arkhipov et al. [67] showed that the heart-shape dimer conformation of the extended two-liganded dimer remained stable. To investigate the effect of the bound-ligand, one and both ligands were removed from the back-to-back structure (Figure 3b). The one-liganded back-to-back conformation is not stable but becomes so when both ligands are removed. The results showed a significant rearrangement; specifically, the gap between DI and DIII left by the detached ligand was filled, both domains coming into contact with each other. There is also a rearrangement of DI and DIII resulting in a bending motion of DIV around the hinge region with DIII leading to an increased separation between the C-terminal portions of the two DIVs. The latter proved to be crucial in providing an understanding of the type of link across the membrane established by these ligand-free inactive complexes (discussed further in Section 4.5). 

In subsequent work, Arkhipov et al. [45] performed MD simulations of the ECM dimer in which each monomer was glycosylated in 10 of the 12 potential Asn sites using three types of glycans (BiS1F1, Man6 and Man8) (Figure 3c). One of the aims was to investigate whether glycosylation could provide any structural basis to the unexplained negative cooperativity in the ligand binding ubiquitously observed for full-length EGFR on cells (see for example [172]). In the context of a monomer-dimer transition, the negative cooperativity of ligand-binding reveals that when one ligand-binding site of EGFR dimer is occupied, the binding of the second ligand takes place with lower affinity [172]. This phenomenon was structurally explained for the EGFR dimer of labelled *D. melanogaster* by the two structurally distinct ligand-binding sites observed in the crystal structure [173], but cannot be explained by the structure of the human back-to-back EGFR dimer, which displays two virtually identical ligand-binding sites [31,83]. Arkhipov et al. showed that negative cooperativity could not be accounted for by glycosylation. However, simulations suggested that one of the EGF molecules bound to the dimer may occasionally rest on the membrane, an event found to be independent of glycosylation, and that the interaction between the EGF ligand and the membrane may lead to a breaking of the symmetry between the two ligands, thus contributing to the negative cooperativity of EGFR ligand binding. We will return to the subject of negative cooperativity in Section 5 in the context of higher order oligomers. 

### 4.2. Active Kinase Dimers and Structural Cross-Talk Across the Asymmetric Interface

For the kinase domain of EGFR to become catalytically active in response to ligand binding, ligand-induced dimerisation must be transduced across the plasma membrane. In a seminal work, Kuriyan and co-workers unravelled the structural details of how dimerisation underpins kinase activation [36] (Figure 4). They discovered that two EGFR kinase domains interact in an asymmetric fashion by which the C-lobe of the ‘activator’ contacts the N-lobe of the ‘receiver’ at points of the αC-helix, the β4/β5 loop and an N-terminal extension of the N-lobe (Figure 4a). Formation of this asymmetric kinase dimer induces allosteric changes in the N-lobe extension of the receiver kinase leading to the conformational changes in its αC-helix and A-loop required to switch on the activated state (Figure 4b). The intimation that brought to light this dimeric mechanism of kinase activation was hidden in the original structure of the EGFR kinase bound with erlotinib [120], which in the crystal lattice showed a sizeable interface between the N-lobe of one kinase domain and the C-lobe of the other. The asymmetric dimer of kinase domains and its allosteric activation mechanism in trans resembles the mechanism by which cyclin-dependent kinase is activated by cyclin, with the C-lobe of the activator kinase domain playing the role of the cyclin. Notably, in the Src/CDK-like inactive conformation the contact points in the hydrophobic surface on the N-lobe of the receiver kinase are sequestered (reviewed in [174]), making them specific to the active conformation. 

Normal functioning of the asymmetric kinase dimer requires kinase domains to cycle between active and inactive states. Because of this cyclic nature, it has become increasingly clear that the function of the receptor can only be understood by taking dynamic properties into account. MD simulations played a leading role in this context. Within the kinase domain, a ‘block-based’ intermediate model was proposed in which sparse clusters of closely interacting residues (e.g., motifs) can maintain a weak association to other motifs and thus pass information between more distant regions of a protein, thereby forming an allosteric network that can orchestrate cooperative protein motions and transmit allosteric signals to distal sites [175]. Dixit and Verkhiver [176] used MD simulations to test the possibility of long-range allosteric communication in the ABL and EGFR kinase monomers and in the EGFR asymmetric kinase dimer. The results provided evidence for a dynamic network of inter-communicating clusters of residues, involving the conserved conformationally adaptive αC-helix in the N-lobe and αI and αF in the C-lobe, which may control long-range inter-domain coupling, thereby allosterically regulating the activation of EGFR’s kinase. It was also found that the T766M gatekeeper mutation enhances long-range communication among allosterically coupled motifs, resulting in the stabilisation of the active kinase form. 

Another example of structural cross-talk between the catalytic site and the asymmetric dimer interface was provided by MD simulations carried out by Shan et al. [164]. These suggested that the L834R mutation increases the order of the N-lobe surface in the monomer, thereby lowering the energetic cost of asymmetric kinase dimerisation in the mutant EGFR compared with wild type. This mechanism was subsequently validated by Red-Brewer et al. [156]. First, a crystal structure of the L834R/T766M double mutant in the active conformation revealed an asymmetric dimer interface essentially identical to that in wild type EGFR. Secondly, the lower energetic costs of kinase dimerisation provided by the pre-ordering of the N-lobe of the mutant monomer are consistent with directional ‘super-acceptor activity’, which is particularly prominent in the TKI-resistant L834R/T766M double mutant. 

### 4.3. Active and Inactive Kinase Dimers and Structural Cross-Talk with the TM, JM and C-Terminal Tail Domains

Crucial evidence for the involvement of the JM domain in the formation of the asymmetric kinase dimer was provided by Jura et al. [177], who compared the asymmetric dimer structure of the HER4 kinase [178], which included the JMB portion of the JM segment, with the asymmetric dimer structure of EGFR, which did not. The realisation was that in the crystal lattice of the HER4 structure, the JMB segment, provided by the receiver kinase domain, latches the two kinase domains together by running along the surface of the C-lobe of the activator kinase, thereby reinforcing dimerisation. Crucially, mutations in conserved C-lobe residues that anchor the JMB latch (e.g., Asn 972, Arg 949, Asp 950, Arg 953 in the C-lobe) were found to inhibit EGFR autophosphorylation in cell-based assays substantially. Coetaneous work by Red-Brewer et al. [179] revealed two crystal structures of the EGFR kinase that extended from residue 645 to 998, thus including the entire inner JM region and some 40 residues of the C-terminus. In these, otherwise canonical, asymmetric EGFR kinase dimer structures the contacts between the donor and receiver kinases include parts of the JM domain (Figure 5a). These showed that, as predicted by Jura et al. [177], the C-terminal half of the JM of the receiver cradles the C-lobe of the activator, thereby stabilising the asymmetric dimer. Interestingly, this work also established that the oncogenic activation of the L834R kinase was also regulated by the JM domain, presumably by aiding further the stabilising effect of the L834R mutation on the asymmetric kinase dimer, as was later shown by a combination of MD simulations, X-ray crystallography and cell-based assays [156,164].

The crystal structure in Figure 5a still misses the majority of the C-terminal tail, which in human EGFR spans ~230 residues (Gln958 to Ala1186) and accounts for ~20% of the receptor mass. This is because the sequence of the tail, being inconsistent with regular secondary structure, is likely to be unstructured and cannot be readily crystallised. Only the kinase proximal C-terminal tail segment has been visualised in crystal structures. Examples include the Src/CDK-like inactive structures of the wild type kinase domain [125] or with the V924R or I682Q mutations, which block the formation of the asymmetric kinase dimer [177] and [180]. These structures display a symmetric arrangement of two kinases interacting head-to-head by their N-lobes with a pseudo two-fold screw symmetry (Figure 5b) in which the first portion of the C-terminal tail (Ser967-Met978) forms an α-helix known as the AP2 helix because it also interacts with the AP2 clathrin adaptor upon phosphorylation of Tyr 974 [181]. In these symmetric structures, the C-terminal tail establishes three interactions with the kinase, all of which are consistent with autoinhibition. In one, the AP2 helix of each kinase mediates the formation of the symmetric interface by interacting with the N-lobe of the partner kinase. The region following the AP2 helix (spanning residues 979–990) is termed the electrostatic hook and contains several acidic residues that interact with the hinge region of the kinase domain. Residues 991–998 form a β-strand that tracks the surface of the kinase domain in a manner that resembles the latch formed by the JM segment preventing the formation of the JM latch that is necessary for activation. Interestingly, long-range communication was impaired in the symmetric form of the kinase dimer [176].

Further information on interactions between the kinase domain and the C-terminus was provided by Gajiwala et al. [161], who revealed six crystal structures of monomeric L834R and L834R/T766-EGFR mutants spanning residues 672–998, which include the kinase domain and part of the C-terminal tail partially ordered. Supported by biochemical and biophysical data, the results showed conformational states of mutant kinase domains and show that both type I and type II inhibitors can recognise the active state of the kinase. Interestingly, the crystal-packing interactions for L834R/T766M kinase domain in complex with the MIG6 inhibitor peptide provided a model for recognition of one of the C-terminal autophosphorylation sites, Tyr-1016, which does not require the receptor to be in a fully activated state. These structures showed an autophosphorylation site interacting with the highly conserved is-Arg-Asp motif from the catalytic loop of the neighbouring molecule, which may be suggestive of the potential mechanism of autophosphorylation in trans. 

It should be noted, however, that the regulatory role played by the C-terminal tail is not yet fully understood. The short helix that contains Tyr992 found in some inactive structures (e.g., [36]) is observed slightly shifted in many active structures (reviewed in [122]). Furthermore, in the catalytically competent asymmetric dimer, a steric clash between the JMB segments and the C-terminal tail suggest that the inhibitory role of the C-terminal tail may only be relevant to the receiver half of the active dimer [122]. What is apparent is that both the JM and C-terminal tail play crucial regulatory roles. This is recapitulated by MD simulations carried out by Mustafa at al. [182], which suggested highly correlated regulatory motions between the ATP-binding kinase core and the flexible JM and C-terminal tail domains as the enzyme cycles between active and inactive states. These results suggest that ATP and substrate binding is allosterically coordinated with kinase dimerisation via conformational changes in the JM and the C-terminal tail. Specifically, this work suggested that residues 980–994 of the C-terminal tail, which interacts with the N-lobe region of the kinase core, function as a ‘fulcrum’ for inter-lobe pivoting by interacting with the hinge regions of the kinase core. 

### 4.4. Structural Coupling Across the Plasma Membrane in Ligand-Bound Active Dimers

The ligand-induced EGFR monomer-dimer transition includes aspects of a sequential model in which a flexible link across the plasma membrane could in principle be sufficient to facilitate the formation of the asymmetric kinase dimer [171]. Structural coupling across the plasma membrane had to be invoked nevertheless to explain other EGFR signalling features, including why intracellular domains, and/or residues, like Thr654, can influence extracellular ligand-binding properties, including negative cooperativity [183], or why different ligands can elicit different signalling responses [9,184]. Results summarised below, demonstrated the presence of structural coupling in ligand-bound dimers.

First, the motif of hydrophobic residues (L_655_RRLL_659_) in EGFR’s JMA suggested an α-helical structure [177], as previously indicated by an NMR structure of a micelle-bound peptide containing the JM segment of EGFR [185]. Based on this, Jura et al. [177] considered models of parallel and anti-parallel helical dimers of the JMA and carried out a mutational study that suggested that the antiparallel dimer was energetically more favourable. Furthermore, the antiparallel JMA dimer could be structurally coupled to an N-crossing dimer arrangement of the TM helices. This model was since validated by NMR using a fragment of EGFR spanning the TM helix and the first 29 residues of the JM segment (residues 618–673) reconstituted in lipid bicelles [186]. Noticeably, a near-full-length receptor model could be built in which the N-terminal part of the N-crossing TM-dimer could be linked across the lipid bilayer to the back-to-back dimer. This was the first model in which the ECM and ICM met across the membrane, with the only obvious gap being the short outer JM segment (Figure 6a).

Endres et al. [186] used NMR to recapitulate the previous results from Jura et al. [177]. From these results, they were able to firmly propose conformational coupling across the plasma membrane in the ligand-bound active dimer. Such coupling indeed occurred via the formation of an N-crossing TM dimer linked to an antiparallel JMA helix dimer, as previously proposed by Jura et al. [177]. Accompanying long timescale simulations by Arkhipov et al. [67] validated these results in an impressive simulation of the near-full-length dimer embedded in a POPC/POPS bilayer (Figure 6b). As discussed above (Figure 3c), the extracellular portion of the dimer was observed to adopt tilted orientations, on occasion making contact with the membrane.

### 4.5. Structural Coupling Across the Plasma Membrane in Ligand-Free Inactive Dimers

Evidence for ligand-free dimers has accumulated over the years, but their role in the cell is not yet completely understood [187,188,189,190,191,192,193,194,195]. Arkhipov et al. [67] assembled a complete version of an inactive dimer that included the simulation generated for the ECM dimer in the absence of bound ligands. In this the larger separation between the C-terminal regions of the two DIVs allowed the two ECMs to be connected to their ICMs via a C-crossing TM dimer (Figure 6c). By controlling the polarization properties and water permeability of the micelles into which EGFR TM domains are resuspended, NMR experimentally revealed a weakly polar C-terminal crossing TM dimer [196]. In turn, coarse-grained simulations involving metadynamics free energy calculations also independently validated the existence of the C-crossing TM dimer [197].

The simulation of the near-full-length inactive EGFR dimer also included a bilayer embedded JMA, connected by the extended JMB to the inactive symmetric kinase dimer, as previously shown by crystallographic data [177] (Figure 5b). Subsequent multiscale MD simulations further revealed that strong interactions between the basic residues in JMA and PIP_2_-containing lipid bilayers aid stabilisation of JMA dimer away from the membrane, thereby promoting a conformation corresponding to an asymmetric kinase domain [198]. Unlike the simulated active dimer (Figure 6b), in which the active core and substrate binding site of the receiver face the cell, in the simulated ligand-free inactive dimer the positively charged patches of the kinase subunits of the symmetric kinase dimer were oriented facing the negatively charged lipid bilayer, which was observed to sequestered them, occluding the substrate-binding sites (Figure 4c). 

### 4.6. Autoinhibition Mechanisms in Ligand-Free Dimers

A comparison between the simulations in Figure 6b,c provides a plausible explanation for how the ligand-free extended dimer can remain autoinhibited in the absence of bound ligand. These simulations suggested that dimer autoinhibition depends on conformational coupling across the plasma membrane, proposing a model of activation by which, upon ligand binding, the TM domains would rotate or twist on the plane of the cell membrane, evolving from a C-crossing dimer into an N-crossing TM dimer. This would reorient the intracellular kinase domain dimer from a symmetric inactive configuration into an asymmetric active form, the so call “rotation model” [199]. However, while the intracellular shape of the simulated symmetric dimer was consistent with low-resolution TEM pictures of the near-complete receptor [171], the extended ECM dimer conformation was incompatible with the stalk-to-stalk, non-extended, kinase-mediated dimer suggested by TEM images of purified, near-full-length EGFR [96]. Furthermore, the sizeable DIV–DIV separation in the one-liganded extended dimer (Figure 3b) would preclude the formation of the N-crossing TM dimer, which is obligatory to form the active asymmetric kinase dimer. This is inconsistent with results from cell-based assays that showed that a single ligand is sufficient to form active EGFR dimers [200]. 

Other possible models of ligand-free dimers include those displayed in Figure 6d,e. Zanetti-Domingues et al. [50] used a repertoire of high-resolution imaging methods, including FLImP, FRET, and single particle tracking, and combined these with atomistic MD simulations to investigate the nature of ligand-free inactive dimers on the cell surface. Contrary to the proposed role of the symmetric kinase dimer in the autoinhibition of wild type EGFR and the TM dimer rotation model of activation [199], the data from Zanetti-Domingues et al. [50] showed that kinase domains of ligand-free, non-monomer wild type EGFR complexes do not interact. Only mutant kinase species bear populations of kinase dimers. A dimer consistent with a ligand-free extended architecture (Figure 6c) was found for L680N-EGFR, a mutant in which the symmetric kinase dimer was stabilised by their inability to form the asymmetric moiety [36]. In turn, binding of type I TKI, removal of plasma membrane cholesterol, both of which promote the asymmetric kinase dimer [96,201], induced across the plasma membrane the formation of a stalk-to-stalk dimer (Figure 6e). Together, these results suggested conformational coupling across the plasma membrane in ligand-free dimers on cells.

The L834R mutation, predicted by previous simulations to promote the formation of the asymmetric kinase dimer [164], was also found to promote inside-out the stalk-to-stalk dimer architecture (Figure 6e), thereby also consistent with conformational coupling across the plasma membrane. Interestingly FLImP results for T766M-EGFR suggested that this mutant can form similar size populations of both back-to-back extended (Figure 6c) and stalk-to-stalk non-extended dimers (Figure 6e). This finding predicted that, if conformational coupling occurs, counterintuitively, T766M-EGFR, which is constitutively active, should form a significant number of symmetric kinase dimers. To test this prediction, MD simulations of the symmetric kinase dimer were carried out and the free-energy landscapes from the parallel tempering metadynamics (PTmetaD) simulations of Sutto et al. [80] were reanalysed and re-projected as a function of different variables for wild type EGFR and T766M-EGFR. The results from the simulations confirmed the experimental prediction, and indicated that the T766M mutation has a dual effect; on one hand destabilising the asymmetric kinase dimer, and on the other stabilising the symmetric one [50]. In turn, single particle tracking experiments showed that the asymmetric and symmetric kinase dimers co-exist in equilibrium at the plasma membrane under the modulation of the C-terminal domain. These results are a further example of the power of combining high-resolution experimental results from cells with atomistic MD simulations. 

## 5. Ligand-Free and Ligand-Bound Multimeric EGFR Assemblies

The presence of ligand-bound and ligand-free higher order EGFR oligomers on the cell surface has been anticipated for years, but the lack of methods with sufficient resolution has hindered progress towards determining the oligomer architecture [202]. This is compounded by the challenge poised to MD simulations in handling extremely large simulations involving several millions of atoms. Towards understanding oligomer architecture, the Kuriyan lab combined a mutational study with the quantitative properties of single-molecule imaging, which can report the number of receptor units in individual cell surface receptor complexes [203], and molecular modelling to generate a model of a ligand-bound multimer [204]. In the resulting model (Figure 7a) the ligand-bound dimers are stacked side-to-side, with a DIV leg from one dimer interacting with a DIV leg from the other. Because these dimers are open structures, higher-order multimers can readily be formed by adding additional dimers through the propagation of the DIV-DIV interaction.

Soon afterward, Needham et al. [49] reported a long-timescale atomic-resolution model of a near full-length ligand-bound tetramer simulated using Anton-based methods. In this tetramer, back-to-back dimers assemble oligomers via unoccupied ligand-binding sites, establishing a ‘face-to-face’ interaction (Figure 7b). The model is also open-ended, growing by the lateral addition of back-to-back dimers joined by face-to-face interfaces. Because the face-to-face interaction largely overlaps with the ligand-binding interface, the model implies that ligand binding and the face-to-face dimer interaction between back-to-back dimers compete with one another. This may be an important factor contributing to the negative cooperativity of EGFR ligand binding. 

The 2D lateral separations between the receptors predicted by the above back-to-back/face-to-face oligomer assembly were experimentally validated using FLImP data at 4.8 nm resolution by Needham et al. [49], whilst FRET data was used to validate the predicted vertical separations between bound ligands to the plasma membrane, which are different in dimers and oligomers. This is another example of the opportunities provided by high-resolution data in being able to constrain atomic-resolution structure when crystallographic data are not available, as it is the case for EGFR oligomers. This work also showed that the role of these oligomers is to organise kinase-active dimers in ways optimal for auto-phosphorylation (Figure 7b). 

Ligand-free oligomers have also been frequently reported (e.g., [49,189,192,193]. To investigate the nature of these oligomers, Zanetti-Domingues et al. [50] revealed a polymer chain architecture by using a combination of high-resolution imaging methods and MD simulations. After the polymer architecture was first hinted by the imaging, monomer crystal structures were searched to find lattice contacts that might reveal previously unidentified oligomer interfaces consistent with this. A suitable one was found in the monomer contacts of the tethered structure bound to the 9G8 nanobody [89] (Figure 7c). Simulations carried out starting from the asymmetric dimer seen in the crystal packing revealed a stable head-to-head dimer conformation (Figure 7d). This was akin to the crystal dimer conformation in that it remains open-ended and asymmetric, maintaining the trans interaction between the DIII of one monomer and the DIV of the other, but it additionally bears a trans interaction between DI and DII. As previously found in the simulations of the ECM of EGFR [67], the tethered conformation of the crystal structure was also consistently unstable. The DI and DIII in both monomers gained stable cis interaction with one another, giving rise to a conformation similar to that in the back-to-back inactive dimer in Figure 6c in terms of DI, DII, and DIII. The separations between DI and DIII to the membrane predicted by the head-to-head dimer model were validated by FRET results. In turn, the biological relevance of the ligand-free head-to-head dimer was suggested by the finding that the amino acids involved in the head-to-head interface (6–238) are missing in the constitutively active variant EGFRvIII prevalent in glioblastoma [205]. 

From the head-to-head dimer, higher order polymers of various lengths were assembled by incorporating additional protomers and repeating the head-to-head interaction [50]. The resulting polymer chain displays curvature in the plane of the membrane that arises because the DIII–DIII separation between nearest neighbours is larger than DI–DI (illustrated in Figure 7e). The resulting curvature is commensurate with the diameter around relevant plasma membrane vesicles [206].

Furthermore, single particle tracking showed that the kinase domains of head-to-head dimers and oligomers do not interact (Figure 7f). Together, this work suggested that the role of the head-to-head interaction is two-fold. On the one hand, it drives oligomer assembly, priming receptors for a quick response to growth factor. On the other, it maintains TM helices sufficiently apart to oppose the formation of kinase-mediated dimers. Indeed, the work also showed that kinase-mediated dimerisation breaks the head-to-head interaction, suggesting the latter is important for autoinhibition.

## 6. Conclusions

In this review, we have brought together sources of structural information from different disciplines regarding the atomistic structure of EGFR in its different oligomer forms and the presence and absence of the ligand stimulus. While it remains indisputable that the depth of our understanding of structure/function relationships is absolutely dependent on the availability of high-resolution structures, the bulk of which in the EGFR field arises from X-ray protein crystallography, it is also increasingly obvious that protein crystallography alone cannot provide crucial details on the functioning of the receptor on the cell surface. This is precisely the environment where anti-cancer drugs are bound to interact with the receptor. Among other challenges, the flexibility of EGFR compounded with its high degree of post-translational modifications, the two-dimensional constraints imposed by the plasma membrane, and interactions between the receptor with other lipids and many proteins, make a multi-disciplinary approach to structure determination a key factor on our success to understand the biology of EGFR signalling. In light of the results reviewed here, we propose that MD simulations are bound to play an increasingly important role in our understanding of EGFR signalling and its dysregulation in oncogenic diseases. Uniquely, in a discipline that investigate mechanisms underpinning the functioning of very complex systems, like biology, atomistic MD simulations can play a unique predictive role, fulfilling, at least in part, the role theoretical principles play in other disciplines, like in physics. To quote from Albert Einstein, in physics “A theory can be proved by experiment; but no path leads from experiment to the birth of a theory”. The development of ever-increasing resolution multi-disciplinary approaches, crucially in cells, in parallel with more powerful MD simulations will be paramount in unleashing the potential of a combined theoretical and experimental approach in biology.

The combination of theory and experiments has already shed much light on the mechanisms by which NSCLC mutations dysregulate the EGFR and modulate its sensitivity and resistance to TKIs. These mutations allow the enzyme to escape the quiescent intrinsically disordered states and increase its time of residence in the catalytically activated state, by facilitating dimerisation of the enzyme (e.g., [80,156,161,164]). Taken the above results together, dysregulated EGFR mutant enzymes may drive NSCLC via changes in their dynamic nature. The greater propensity to form asymmetric dimers is a ‘game changer’ in the ligand-free state, bringing about the formation of constitutively active kinase-mediated dimers that can bypass the autoinhibition provided in the absence of ligand by extracellular interactions [50], thereby increasing the activity further. 

The ability of cell surface receptors to form higher order oligomers has been hailed as a new paradigm in signal transduction [207]. The nature of higher order assemblies must be understood in detail because their presence can explain functional features that cannot be accounted in the context of dimers, like, for example, threshold behaviour, signal amplification and biological noise reduction [208,209]. However, the overall role of oligomerisation in signal transduction, and the mechanisms driving oligomer formation have not yet been resolved. Delineating an in cellulo structural ‘solution’ to these properties exploiting somatic mutations and novel approaches will profoundly impact our understanding of growth factor receptor signalling via higher-order assemblies and guide novel opportunities for therapeutic intervention. The combination of MD simulations with super-resolution methods like FLImP, single particle imaging and tracking, and FRET is beginning to make substantive inroads towards this goal.

## Figures and Tables

**Figure 1 cells-08-00316-f001:**
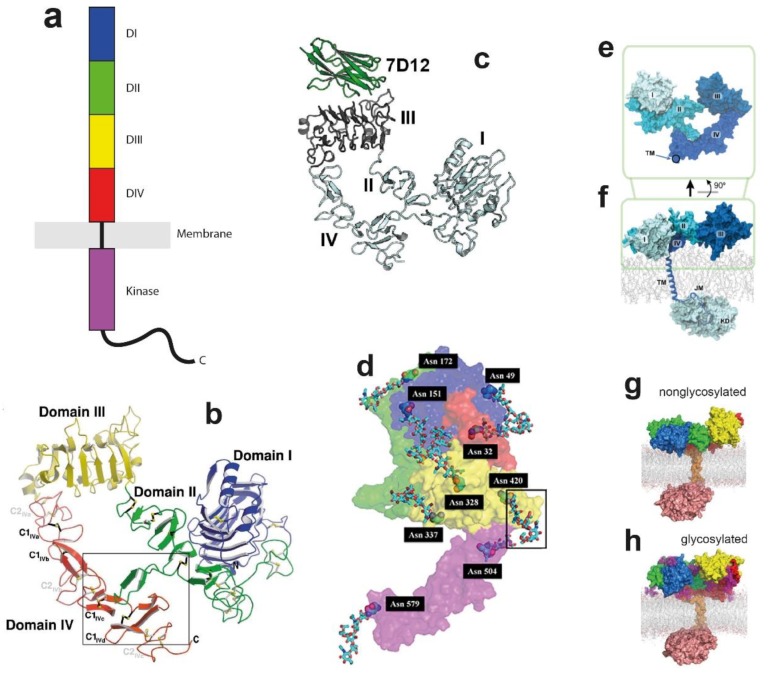
Topology, structure, and simulations of the epidermal growth factor receptor (EGFR) monomer. (**a**) Cartoon of an EGFR monomer, showing its domain structure. (**b**) Ribbon representation of the solubilised monomeric extracellular module (ECM) of EGFR (sEGFR) structure (Protein Data Bank (PDB): 1NQL), showing the “tether”, in which an α-hairpin from DII forms an interaction with DIV. Disulfide bonds are labelled in black (C1) and gray (C2). Taken, with permission, from Ferguson et al. [85]. (**c**) Ribbon representation of sEGFR with bound VHH domain 7D12 (PDB: 4KRL), again showing the presence of the tether. Taken, with permission, from Schmitz et al. [89]]. (**d**) Extended monomer structure of EGFR (PDB: 3NJP), taking a monomer unit from the ligand-bound back-to-back dimer. The locations of glycosylated Asparagine residues are shown, one highlighted inside a grey box. Taken, with permission, from Irani [93]. (**e**) A molecular dynamic (MD) simulation of the sEGFR started from the structure in d. The connecting point between the extracellular and the transmembrane helices is marked by a circle. (**f**) As (**e**) but showing the near-complete receptor. Taken, with permission, from Arkhipov et al. [67]. (**g**,**h**) Endpoint structures from 1 µs simulations of the (**g**) nonglycosylated and (**h**) glycosylated EGFR in 1,2-dioleoylsn-glycero-3-phosphocholine (DOPC)/ sphingomyelin (SM)/cholesterol membranes. Subdomain DI shown in blue, DII in green, DIII in yellow and DIV in red. The single-pass transmembrane (TM) domain is shown in orange, the intracellular tyrosine kinase domain (TKD) in salmon, and the glycans in purple. Taken, with permission, from Kaszuba et al. [94].

**Figure 2 cells-08-00316-f002:**
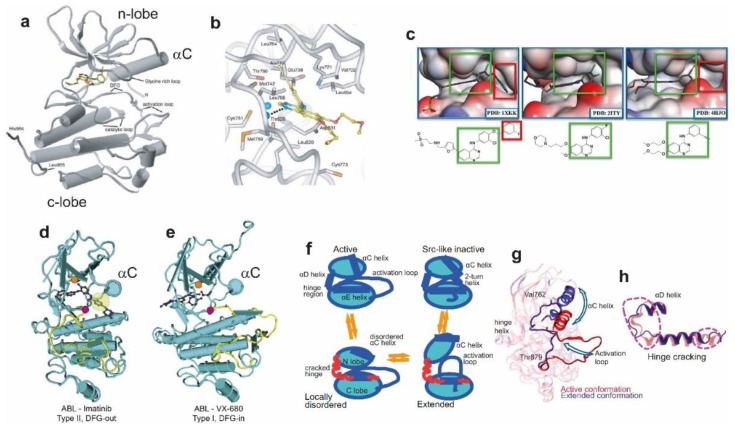
Structure and dynamics of the kinase domain. (**a**) Structure of the EGFR kinase domain with the inhibitor erlotinib bound in the cleft between the amino-terminal and carboxy-terminal lobes (PDB: 1M14). Taken, with permission, from Stamos et al. [120]. (**b**) The inhibitor binding site and nearby residues from the EGFR kinase domain complexed with erlotinib. The dashed line indicates an H-bond from the receptor to the drug, and a water molecule is shown as a pale blue sphere. Taken, with permission, from Stamos et al. [120]. (**c**) Structures of the EGFR binding site with three different inhibitors bound; 1XKK (Lapatinib) [125], 2ITY (gefitinib) [126], 4HJO (erlotinib) [127]. (**d**) Structure of Abelson murine leukemia viral oncogene homolog 1 (ABL) kinase domain complexed with the type II inhibitor imatinib, showing the DFG-out conformation [128]. Both panels were taken, with permission, from Treiber and Shah [129]. (**e**) Structure of ABL kinase domain complexed with the type I inhibitor VX-680, showing the DFG-in conformation [130]. (**f**) MD simulations of the EGFR kinase monomer in the time scale of 25 µs showing transitions between the active state and both the Src-like inactive and the DFG-out inactive states of EGFR kinase. (**g**) Comparison of the extended conformation (blue) with the initial active conformation (red). (**h**) Close-up of the hinge region, with and without hinge cracking (blue and red, respectively). (**f**–**h**) taken, with permission, from Shan et al. [131].

**Figure 3 cells-08-00316-f003:**
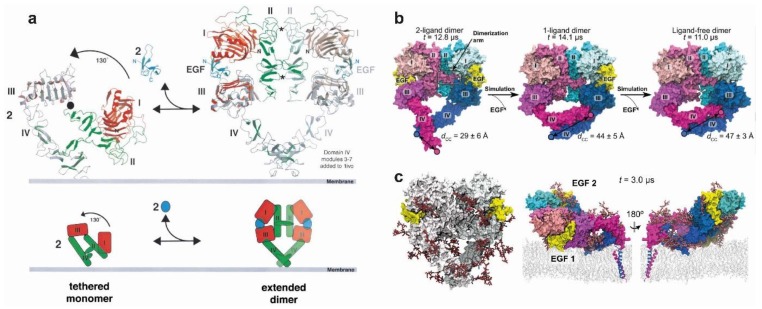
The ligand-bound extracellular back-to-back dimer. (**a**) A model of ligand-induced EGFR dimerization. The transition is shown between the monomer tethered structure on the left (model derived from [85]) and the active dimer on the right (model derived from [31]). The “back-to-back” dimer includes the DIV modules missing from earlier structures. Taken, with permission, from Burgess et al. [170]. (**b**) Simulation of the effects of removal of the ligand, showing a significant rearrangement by which the gap between DI and DIII left by the detached ligand is filled, and a bending motion of DIV around the hinge region with DIII leads to a separation between the C-terminal portions of DIV. Taken, with permission, from Arkhipov et al. [67]. (**c**) A simulation showing the fully glycosylated ectodomain dimer. The glycans are coloured by atom type (carbon in grey, oxygen in red and nitrogen in blue). The conformation after 3 µs of simulation is shown on the right, from two opposing viewpoints. Taken, with permission, from Arkhipov et al. [45].

**Figure 4 cells-08-00316-f004:**
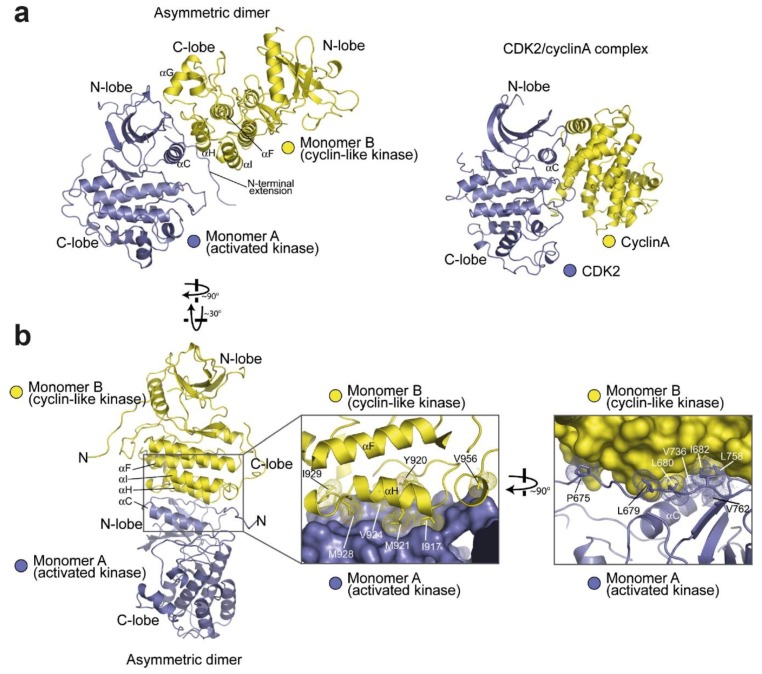
Structure of the asymmetric crystallographic dimer of the EGFR kinase domain (taken, with permission, from Zhang et al. [36]). (**a**) Shows an overview of the dimer (PDB:2GS6) compared with the cyclin-dependent kinase-2 (CDK2)/cyclin A complex (PDB: 1HCL). (**b**) Shows detail of the dimer interface. The first zoomed-in area (left) shows a ribbon structure for Monomer B with its hydrophobic interface residues highlighted. The second zoomed-in area (right) shows the reverse, with the hydrophobic interface residues of monomer A highlighted.

**Figure 5 cells-08-00316-f005:**
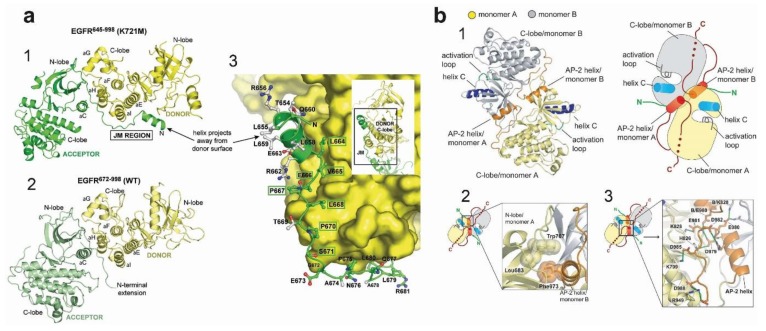
Asymmetric and symmetric kinase dimers showing interactions with the juxtamembrane (JM) and C-terminal domains. (**a**) Crystal structure of the EGFR kinase from residue 645 to 998, including the entire inner JM region and some 40 residues of the C-terminus. The structure shows that the C-terminal half of the JM of the receiver cradles the C-lobe of the activator, thereby stabilising the asymmetric dimer. (1) shows the structure with an inactivating K721M mutation, closely resembling the structure for EGFR lacking the JM region (2). (3) shows the detail of the side-chains in the acceptor JM region (green), in contact with the C-lobe of the donor kinase domain. Taken, with permission, from Red-Brewer et al. [179]. (**b**) Structure of the symmetric inactive dimer. (1) gives an overview of the structure, while (2) shows the hydrophobic packing between the C-terminal AP-2 helix and the N-lobe of monomers A and B, respectively. (3) shows the electrostatic hook that forms between the C-terminal tail of EGFR and the hinge region of its kinase domain. Taken, with permission, from Jura et al. [177].

**Figure 6 cells-08-00316-f006:**
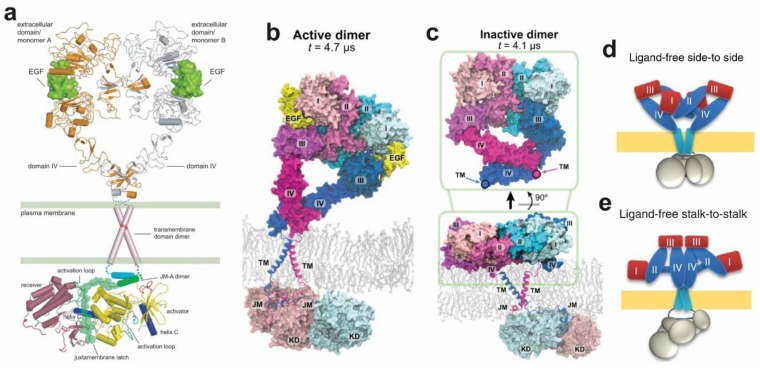
(**a**) Full EGFR model in which the ECM and intracellular module (ICM) meet across the membrane, the only gap being the short outer JM segment. Taken, with permission, from Jura et al. [177]. (**b**,**c**) simulation of the near full-length active and inactive dimers, respectively. Taken, with permission, from Arkhipov et al. [67]. (**d**,**e**) models for alternative dimers produced by a combination of high-resolution imaging methods, including fluorescence localization imaging with photobleaching (FLImP), Förster resonance energy transfer (FRET), and single particle tracking and atomistic MD simulations. Taken, with permission, from Zanetti-Domingues et al. [50].

**Figure 7 cells-08-00316-f007:**
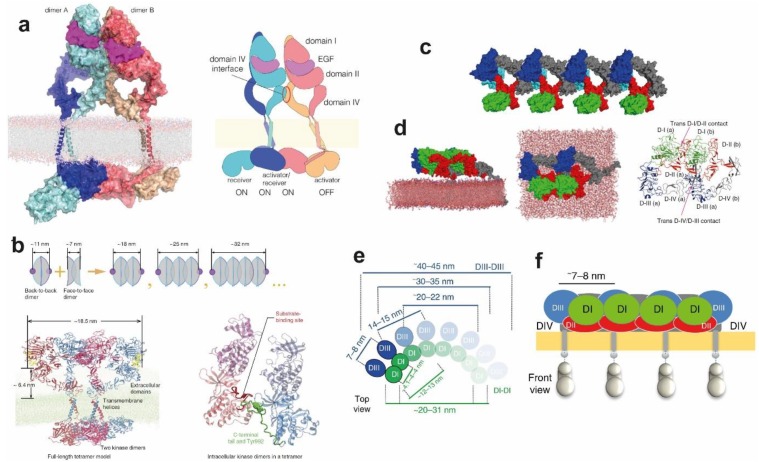
Structure and architecture of EGFR multimers. (**a**) A model for an EGFR tetramer, generated by connecting extracellular modules (PDB: 3NJP) to the structure of dimeric transmembrane helices (PDB: 2M20) and a chain of kinase domains (PDB: 2GS6 and 3GOP). Taken, with permission, from Huang et al. [204]. (**b**) A structural model for EGFR tetramers derived from FLImP measurements and MD simulation. The figure shows an illustration of an open-ended oligomerization scheme for EGFR extracellular domains based on repeating the back-to-back and face-to-face interactions. The two structures below show the full-length tetramer model and the arrangement of the two intracellular active kinase dimers in that model. The phosphorylation site Tyr992 (green) of one receptor is positioned in the proximity of the active site (red) of its neighbour’s kinase domain. Taken, with permission, from Needham et al. [49]. (**c**) A model of an open-ended oligomer 9G8-bound EGFR extracellular domains in the inactive conformation. This was assembled from crystal contacts in PDB ID 4KRP42. 9G8-NB is shown in cyan, EGFR DI in green, DII in red, DIII in blue, and DIV in grey. (**d**) A model dimer structure of free EGFR extracellular domains and their TM domains in the lipid bilayer (based on a snapshot of a simulation at 20 µs). In the left and middle panels, one of the two transmembrane helices can be seen. On the right is shown the dimer seen from the membrane, highlighting the interaction between DI and DII and between DIV and DIII. (**e**) The architecture of ligand-free head-to-head polymers, showing a polymer chain formed by repeating the head-to-head interface, based on separations derived from FLImP. The intensity is graded according to the abundance of the particular oligomer size. (**f**) A ligand-free oligomer viewed from the front, showing the separation between non-interacting ICM units predicted by extracellular head-to-head interactions. (**c**–**f**) taken, with permission, from Zanetti-Domingues et al. [50].

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
