# Peer review of "Structure and Dynamics of the EGF Receptor as Revealed by Experiments and Simulations and Its Relevance to Non-Small Cell Lung Cancer"

_cells, 2019, doi:10.3390/cells8040316_

Round 1

Reviewer 1 Report

The paper by Gervasio and coworkers offers a great perspective on EGFR, its structural and dynamic organization and their relevance for cancer.

The paper is well written and easy to read. I think it provides a comprehensive wrap of data, studies and functional analysis on this important protein.

As a minor revision, the authors may briefly discuss the relevance of EGFR interactions with chaperones systems, such as those discussed in 10.1021/acs.jmedchem.8b00825.

In the dynamics sections they may consider including references to methods that have studied the influence of mutations on the internal dynamics of the proteins in relation to their funcitonal activation, such as in 10.1021/acs.jctc.7b00997, 10.1021/acs.jcim.5b00270 and others.

Author Response

Comment 1: As a minor revision, the authors may briefly discuss the relevance of EGFR interactions with chaperones systems, such as those discussed in 10.1021/acs.jmedchem.8b00825. 

We included new paragraph and references on page 2

Comment 2: In the dynamics sections they may consider including references to methods that have studied the influence of mutations on the internal dynamics of the proteins in relation to their funcitonal activation, such as in 10.1021/acs.jctc.7b00997, 10.1021/acs.jcim.5b00270 and others.

We included new paragraph and references on page 14

Reviewer 2 Report

The review titled "Structure and Dynamics of the EGF Receptor as Revealed by Experiments and Simulations and its Relevance to Non-Small Cell Lung Cancer" is a very clear and easily readable attempt to summarize structural and biological properties of epidermal growth factor receptor and its ligands. Nowadays, in NSCLC the importance of EGFR is highly linked to the clinical benefits arising from TKIs treatment from diagnosis to relapse. Overall, the work is of high contribution to the field, it is well organized and comprehensively described. 

Author Response

No reply required